# Retrospective Analysis of Intra-Aortic Balloon Pump Use in Cardiology Ward Patients Undergoing Coronary Angiography between 2012 and 2020

**DOI:** 10.3390/jcm12041567

**Published:** 2023-02-16

**Authors:** Tomasz Bochenek, Patrycja Sowula, Małgorzata Rodak, Anna Rybicka-Musialik, Bartosz Gruchlik, Katarzyna Mizia-Stec

**Affiliations:** 1First Department of Cardiology, School of Medicine in Katowice, Medical University of Silesia, 40055 Katowice, Poland; 2Upper-Silesian Medical Center, 40635 Katowice, Poland

**Keywords:** intra-aortic balloon pump, percutaneous coronary interventions, outcome

## Abstract

We aimed to evaluate the rate and risk factors of in-hospital mortality in patients undergoing coronary angiography/angioplasty with IABP use as support. We included 214 patients (mean age: 67.5 ± 7.5 years, M/F: 143/71) with an IABP used as the periprocedural support between 2012 and 2020. The main indications for an IABP were cardiogenic shock (143 pts; 66.8%: 55 survivors (51.9%)/88 non-survivors (81.5%); *p* < 0.001) and infarction with an initial significant impairment of ventricular function (34 pts; 15.9%: 21 (19.8%)/13 (12%); *p* = 0.12). In-hospital death was the endpoint of this study. In-hospital death occurred in 108 (50.5%, M/F: 69.4%/30.6%) patients. The mean hospitalization time was 7 days (2–13); deaths occurred more frequently on the first day after the procedure (1 (1–3 days) vs. 3 (1–8), *p* < 0.001); and the mean hospitalization time was 2 days (1–6) for non-survivors vs. 11 days (7–17) for survivors (*p* < 0.001). Regarding the patients who did not survive, they were older (69 vs. 66.5, *p* = 0.043), their LVEF was lower (0–15%: 15 (13.9%) vs. 12 (11.3%); 16–40%: 73 (67.6%) vs. 65 (61.3%); >40%: 14 (13%) vs. 29 (27.4%); *p* = 0.007), and hyperlipidemia was less common (30 (27.8%) vs. 55 (51.9%) pts, *p* = 0.001) than in those who survived. The IABP is still a method for cardiac support; however, mortality limits its use.

## 1. Introduction

The intra-aortic balloon pump (IABP) is currently still used in cardiac surgery, cardiology, and intensive care as a device that provides mechanical circulatory support (MCS). Nevertheless, there is a disparity between the recent clinical evidence and the clinical use of this support device. At present, new devices such as Impella and ECMO are increasingly available. Still, despite the new, more advanced technologies, controversies surrounding their effectiveness vs. the rate of complications exist. The ease of use and the common presence of the IABP in hospital wards worldwide lead to the relatively frequent use of this pump compared to other devices. In many hospitals, often those without surgical backup, it still remains one of the main LV support devices in cardiology, especially in sudden critical situations.

The current indications for an intra-aortic balloon pump are myocardial infarction with cardiogenic shock, requiring support in cardiac surgery, and high-risk PCI. Cardiogenic Shock (CS) is related to decreased cardiac output, which justifies the balloon insertion. The European Society of Cardiology’s 2018 Guidelines on myocardial revascularization do not recommend routine use of an IABP in patients with acute myocardial infarction (AMI) that is complicated by CS (class III B); regarding this, one should remember that the word “routine” is crucial [1]. According to the document, an IABP may be considered only in patients with STEMI, particularly conditions such as mechanical complications, i.e., severe mitral insufficiency or ventricular septal defects [2]. The results from the IABP-Shock II Trial are responsible for the changes in these guidelines; this study showed that the IABP had no positive impact on long-term mortality in patients after CS was induced by myocardial infarction [3]. Additionally, the CRISP AMI randomized trial did not confirm the expected assumption of a reduction in the infarct size in the case of the implementation of the IABP with PCI revascularization [4]. The first conclusions of BCIS-1 were not promising, due to differences in the endpoints between the group with the IABP and the group without it [5]. Three years later, Perera et al. published further outcomes that demonstrate reduced long-term mortality in patients supported with an intra-aortic balloon. According to the BCIS-1 results, the counter-pulsation may be used as an assist in high-risk PCI [6,7].

In our study, we wanted to look closer at the rate and risk factors of in-hospital mortality in patients with an IABP undergoing coronary angiography/angioplasty.

## 2. Materials and Methods

We performed a single-center retrospective analysis of patients undergoing coronary exploration (*n* = 214: non-survivors *n* = 108 vs. survivors *n* = 106) with IABP use as periprocedural support. Our center is a high-volume tertiary cardiac center in south Poland, and in the period that we analyzed, there were, on average, 6000 coronary angiography, more than 3000 angioplasties, and 2000 acute coronary syndrome (ACS) admissions a year, with the majority being treated invasively. We analyzed patients who were hospitalized between 2012 and 2020. In this study, we analyzed IABP use in all patients hospitalized in cardiology wards undergoing coronary angiography/angioplasty, not only ACS admissions. We used standard protocols where cardiogenic shock, mechanical complications of acute myocardial infarction (AMI), unsuitable and challenging coronary anatomies, as well as complications of the performed procedures (no reflow, artery occlusion, severe dissections) were the reasons for temporary support. In the analyzed period, the IABP constituted the major LV support in our center. ECMO support was used in the cardiac surgery unit, but those patients were not included in the analysis. 

Two-hundred and fourteen patients (mean age 67.5 ± 7.5) in whom an IABP was used as hemodynamic support during coronary interventions were included in the analysis: 143 men (66.8%) and 71 women (33.2%). They were divided into two groups: patients who survived (survivors) and patients who died during hospitalization (non-survivors). All data were obtained using medical record entries from a computer medical system. Patients who were directly admitted to intensive care requiring an “a priori” invasive ventilation unit were not analyzed in our study, which constitutes a limitation. 

Statistical analysis was performed using the Statistica 13.0 StatSoft program with an assumed significance level of *p* = 0.05. Basic descriptive statistics were calculated: median and interquartile ranges for variables were not characterized by a normal distribution; additionally, the frequency of occurrence was calculated for nominal variables. The relationship between the groups was calculated using the Mann–Whitney U test and Pearson’s chi-squared test. The analyzed model was developed using logistic regression.

## 3. Results

### 3.1. In-Hospital Mortality

Intra-hospital death occurred in 108 patients (50.5%): 75 men (69.4%)/33 women (30.6%) were enrolled in the study. Patients who died were characterized by a higher age. In the survival group, the median age was 66.5 (59–75) vs. 69 (63–79), the *p* = 0.043 in those who died, and a dependence on stenting in one or more coronary arteries was noted.

### 3.2. Clinical Characteristics of the Study Groups: Survival vs. Death 

The mean hospitalization time for all patients was 7 days (2–13). Those patients who survived spent, on average, 11 days in the hospital, and those who died mainly died shortly after admission—on average, this was 2 days (11 (7–17) vs. 2 (1–6); *p* < 0.001). Of all patients, 90 (42.1%) of them required subsequent treatment in the intensive care unit. In the hospital, death occurred in 59 (54.6%) patients who were hospitalized in the intensive care units (*p* < 0.001). A normal sinus rhythm during admission to the hospital was observed in 117 (54.7%) patients (66/62.3% in survivors vs. 51/47.2% in non-survivors, *p* = 0.043 (Table 1)). Atrial fibrillation on admission occurred in 21 (9.8%) patients (8/7.5% in survivors vs. 13/12% in non-survivors, *p* = 0.316).

Comorbidities collected from patient data included the following: obesity; arterial hypertension (HA); diabetes mellitus (DM); coronary artery disease (CAD); hyperlipidemia and peripheral atherosclerosis; a history of myocardial infarction (MI); and past coronary artery bypass graft (CABG) and angioplasty procedures (Table 1). There was no statistical influence on the patients’ survival or death in our data analysis. Unexpectedly, only hyperlipidemia occurred more often in the case of survivors (*p* = 0.0011; survival of 55 patients; death of 30 patients); in the remaining cases, we did not observe statistically significant differences (Table 1). The LVEF parameter in all 208 patients had the following values: 0–15% for 27 patients (12.6%); 16–40 for 138 patients (64.5%); and over 40 for 43 patients (20.1%). No LVEF data were available for six patients (2.8%).

Significantly, patients who survived had lower mean values for the following parameters: creatinine, minimal troponins, maximal CK MB, maximal glycemia, and maximal white blood cells (Table 2). In addition, an increase by one unit of minimal troponin T caused an over 1.34 times higher risk of death (OR 1.34, 95% CI 1.07 to 1.68, *p* = 0.0105).

### 3.3. Indications for IABP in the Study Groups: Survival vs. Death

An IABP was used in all analyzed patients. An IABP was inserted in 8 patients before coronary angiography, in 182 during the procedure, and in 2 after the procedure. For the remaining patients, the timing was difficult to establish. The exact reason for the insertion of the balloon pump during percutaneous intervention was not analyzed in depth for this population. 

No differences were noticed between the patients’ survival and death depending on the timing of the insertion of the counter-pulsation. The IABP running time (survivors vs. non-survivors *p* < 0.001) ranged from 2 days (1–5) in all the patients to 3 days (1–8) for those who survived. Deaths were recorded mainly on the first day after IABP insertion (1–3 days).

The general indications for an IABP are summarized in Table 3. The main indication for an IABP was cardiogenic shock (143 patients; 66.8%), where 55 patients survived (51.9%), and 88 (81.5%), *p* = 0.00, did not survive. This was followed by myocardial infarction with an initial significant impairment of left ventricular function (34 patients; 15.9%), of which 21 (19.8%) patients survived and 13 (12%) died. High-risk PCI (no reflow, supportive care) was performed in 33 (15.4%) subjects out of the total number of patients. A total of 25 (23.6%) of them survived, but death was reported in 8 patients (7.4%). Life-threatening cardiac arrhythmias (VT, VF) occurred in a total of 11 (5.1%) patients (5/4.7% in survivors vs. 6/5.6% in non-survivors). Subsequently, mechanical complications of myocardial infarction involving the mitral valve occurred in 10 (4.7%) patients (5/4.7% in survivors vs. 5/4.6% in non-survivors). A ventricular septal defect occurred in seven (3.3%) patients (1/0.9% in survivors and 6/5.6% in non-survivors).

### 3.4. Non-Fatal Complications after IABP Insertion: Survival vs. Death

Non-fatal complications after IABP insertion were found in a total of 38 patients (17.8%). The most common was limb ischemia, occurring in 15 patients (7%). Then, it was hematoma in 12 patients (5.6%, which was significantly more common among surviving patients than among those who died: 11 vs. 1; *p* < 0.001). Gastrointestinal bleeding was noted in nine patients (4.2%). An aneurysm complication occurred in one patient (0.5%), similar to stroke (0.5%).

### 3.5. Coronary Angiography in the Study Groups: Survival vs. Death

Coronary angiography revealed significant stenosis (above 50%) in the following coronary arteries: left anterior descending artery (LAD; 147/68.7% of patients), circumflex artery (Cx; 88/41.1%), right coronary artery (RCA; 86/40.2%), left main coronary artery (LM; 78/36.4%), obtuse marginal artery (OM; 14/6.5%), and diagonal branch (D; 10/4.7%). Changes in any of these arteries had no impact on the patients’ survival. An identical lack of dependence was noted for stenting one or more coronary arteries. Coronary angioplasty with a stent was performed in LAD (127/59.4%), Cx (60/28%), RCA (44/20.1%), LM (63/29.4%), OM (8/3.7%), and D (9/4.2%). A total of 78 (36.5%) patients had at least two different arteries stented.

### 3.6. Multivariate Analysis: Risk Factors for In-Hospital Mortality 

Table 4 shows the results of estimating the best logit model for the probability of death. The risk of death was almost 2 times (OR 0.43) lower among patients who had hyperlipidemia. In addition, an increase of one unit of troponin resulted in a more than 1.37-fold higher risk of death. The higher the patient’s age, the greater the likelihood of death (OR = 1.03).

## 4. Discussion

The intra-aortic balloon pump (IABP) is still used as mechanical circulatory support for hemodynamic stabilization in patients with heart disease. In addition to being a well-known circulatory support device, the IABP is also the simplest and easiest to implant and explant in the invasive cardiology laboratory. There is a lack of data on the actual use of the intra-aortic balloon pump (IABP) in various cardiogenic shocks (CSs) and its relationship with patient outcomes [8]. In the present study, all patients underwent angiography and/or PCI and had IABP support. Our research shows that the intra-aortic balloon pump procedure is used more often in men than in women. Aging is also an important factor in cardiovascular function deterioration; this factor results in an increased risk of cardiovascular disease (CVD) in the elderly. The risk is potentiated by additional factors, such as frailty, obesity, and diabetes. Studies have shown that the incidence of CVD increases with age in both men and women, including the incidence of atherosclerosis or myocardial infarction [9,10]. In this study, age was also found to be significant. It was shown that the higher the age of the patient, the higher the probability of death (OR = 1.03). Patients who died were characterized by a lower LVEF, most often 16–40%, while in survivors, the LVEF was higher. In the patients, the most common comorbid disease was peripheral arteriosclerosis, followed by arterial hypertension. The 2019 ESC/EAS guidelines for the management of dyslipidemia highlight the strongly proven association of lipid disorders as a risk factor for atherosclerotic cardiovascular disease (ASCVD) [11]. Our results showed that hyperlipidemia could be associated with a higher death rate (OR = 0.43) in patients who underwent PCI and required IABP support. Another study found elevated lipid levels to be an independent predictor of 1-year mortality after PCI [12]. Higher levels of total cholesterol, non-HDL cholesterol, and remnant cholesterol have been linked with MACE occurrence, which is related to mortality in patients with MI [13]. Most often, an IABP is inserted during surgery, but the time of insertion does not affect patients’ survival. In our study, the main reason for introducing an IABP was due to the presence of cardiogenic shock. Very rarely, life-threatening cardiac arrhythmias (VT, VF) occurred.

Despite the use of an IABP, about half of the patients died. They died mainly on the first day after the insertion of the IABP. Our results are consistent with another study where the highest mortality occurred on the same day [14]. There were no differences between patient survival and death according to the timing of counter-pulsation insertion, similar to a study that aimed to evaluate the outcomes of patients undergoing IABP insertion before vs. after primary PCI in acute myocardial infarction that were complicated by cardiogenic shock. The analysis included 275 patients, and the timing of IABP implantation before or after primary PCI did not affect the outcomes in these patients [15]. In large randomized controlled trials in patients with acute coronary syndrome (ACS) and high-risk percutaneous coronary intervention (PCI), there was no benefit of IABPs in reducing the infarct size and no difference in major adverse cardiac and cerebrovascular events (MACCE) between patients undergoing high-risk PCI, either with or without IABP support. There was also no demonstrated benefit in short-term or long-term morbidity or mortality. Despite the promising hemodynamic effects, studies have shown that the IABP does not change mortality outcomes in patients with AMI-CS or complex PCI [16]. However, we noticed a relatively small number of complications related to the insertion of the IABP. The most common was limb ischemia (7% of all subjects). Hemorrhagic incidents, such as hematoma (5.6%) and gastrointestinal bleeding (4.2%) were not frequent. The advantage of the IABP is the low level of bleeding complications. The CRISP AMI randomized trial showed no significant differences between the PCI with IABP group and the PCI-alone group [17]. Additionally, compared to other heart support devices that are used among patients with AMI, there are fewer bleeding complications with the IABP [18,19,20,21]. Higher creatinine levels were reported in the non-survivors, which is similar to the results of other studies where higher creatinine levels were associated with increased mortality [22,23]; thus, they may be used as an independent predictor of short-term all-cause mortality [24]. The cardiac troponin (cTn) concentration is the preferred marker of myocardial necrosis. Studies have shown that elevated blood levels of cardiac troponin (cTn) in myocardial injury show a strong association with an adverse prognosis in patients with acute coronary syndromes [25,26]. In one study involving patients with both type 2 diabetes and stable ischemic heart disease, baseline cardiac troponin T levels above the upper limit of normal were associated with an approximately 2-fold increased risk of myocardial infarction, stroke, heart failure, death from cardiovascular causes, and death from any cause. It was concluded that the concentration of cardiac troponin T was an independent predictor of death from the above-mentioned causes [27]. Increased troponin levels and a worse outcome are supported by the results presented in this study, where a one unit increase in troponin levels results in a more than 1.37-fold increase in the risk of death. A follow-up of this study is needed to assess this.

We are aware of some limitations of the study. We did not assess the degree of shock separately, nor did we analyze the end organ dysfunction makers. This was a retrospective analysis and we had limited access to the all clinical data. Moreover, the analyzed period was long, and some data unfortunately were not available, e.g., lactate levels. The therapy was not guided by invasive measurements for cardiac output.

## 5. Conclusions

Our study summarizes a one-center, 8-year experience of IABP use for support in invasive procedures. We may assume that the IABP further constitutes a method for hemodynamic periprocedural support in patients undergoing invasive coronary angiography; however, high mortality limits its reasonable use.

## Figures and Tables

**Table 1 jcm-12-01567-t001:** Characteristics of the study groups.

Variables	Survivors (*n* = 106)	Non-Survivors (*n* = 108)	*p*
Sex	M 68 (64.1%)/F 38 (35.8%)	M 75 (69.4%)/F 33 (30.6%)	NS
Age (years)	66.5 (59–75)	69 (63-79)	0.043
Sinus rhythm at admission	66 (62.3%)	51 (47.2%)	0.043
Atrial fibrillation at admission	8 (7.5%)	13 (12%)	0.316
LVEF (%)			0.007
0–15	12 (11.3%)	15 (13.9%)	
16–40	65 (61.3%)	73 (67.6%)	
Above 40	29 (27.4%)	14 (13%)	
No data	0 (0.0%)	6 (5.6%)	
Obesity	25 (23.6%)	19 (17.6%)	NS
Hypertension	76 (71.7%)	64 (59.3%)	NS
Diabetes mellitus	43 (40.6%)	43 (39.8%)	NS
Hyperlipidemia	55 (51.9%)	30 (27.8%)	0.001
Peripheral atherosclerosis	25 (23.6%)	37 (34.3%)	NS
Coronary disease	105 (99.1%)	104 (96.3%)	NS
Prior MI	26 (24.5%)	27 (25.0%)	NS
Prior CABG	10 (9.4%)	7 (6.5%)	NS
Prior PTCA	25 (23.6%)	17 (15.7%)	NS

LVEF—left ventricular ejection fraction; MI—myocardial infarction; CABG—coronary artery bypass surgery; and PTCA—percutaneous transluminal coronary angioplasty.

**Table 2 jcm-12-01567-t002:** The laboratory tests.

Variables	Survivors (*n* = 106)	Non-Survivors (*n* = 108)	*p*
CRP max.	76.00 (31.7–171.8)	126.25 (43.95–212.75)	NS
Creatinine max.	1.095 (0.91–1.51)	1.58 (1.11–2.43)	<0.001
Troponin T min.	0.14 (0.04–0.79)	0.48 (0.10–2.34)	0.001
Troponin T max.	1.92 (0.53–5.41)	4.30 (0.49–10)	NS
CK-MB max.	61.5 (28–134)	121 (44–325)	0.002
Glycemia max.	179 (136–250)	222 (161.5–330.5)	0.002
Hb min.	10.55 (8.7–12.5)	10.9 (8.8–13.1)	NS
WBC max.	15.20 (11.29–20.47)	16.69 (13.57–23.12)	0.027
PLT min.	177.5 (125–208)	174 (117–227)	NS

CRP—C-reactive protein; HB—hemoglobin; WBC—white blood cells; and PLT—platelets.

**Table 3 jcm-12-01567-t003:** Indications for an IABP in the study groups.

Variables	Survivors (*n* = 106)	Non-Survivors (*n* = 108)
Cardiogenic shock	55 (51.9%)	88 (81.5%)
Mechanical complications of AMI		
Acute mitral regurgitation due to papillary muscle rupture	5 (4.7%)	5 (4.6%)
Ventricular septal rupture	1 (0.9%)	6 (5.6%)
Myocardial infarction with decreased left ventricular function leading to hypotension	21 (19.8%)	13 (12%)
Prophylaxis or adjunct treatment in high-risk percutaneous coronary intervention	25 (23.6%)	8 (7.4%)
Low cardiac output state after coronary artery bypass grafting surgery	1 (0.9%)	0 (0.0%)

**Table 4 jcm-12-01567-t004:** Estimation results of the best logistic model for the probability of death.

Logistic Model = 62.70; *p* < 0.0001	Estimation Parameter	*p*	Odds Ratio	95% Confidence Interval
Constant term	−3.19	0.01	0.04	0.00
Age	0.03	0.04	1.03	1.00
Hyperlipidemia	−0.83	0.02	0.43	0.22
Troponin T min.	0.31	0.00	1.37	1.10

## Data Availability

No new data were created or analyzed in this study. Data sharing is not applicable to this article.

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
