# Peer review of "Retrospective Analysis of Intra-Aortic Balloon Pump Use in Cardiology Ward Patients Undergoing Coronary Angiography between 2012 and 2020"

_jcm, 2023, doi:10.3390/jcm12041567_

Round 1
Reviewer 1 Report
The authors analyzed the survival rate and mortality of the patients who underwent IABP because of cardiogenic shock occurred by AMI. This kind of retrospective analysis is important. However, the authors should disclose their whole results of AMI in their hospital during study period. They should also show the use of other mechanically support devices for acute heart failure.
What is the application of IABP for AMI parents in the authors’ hospital? Just heart failure? We use IABP for the patients with coronary no-reflow after PCI. How many patients are included in the study?
P2L18: in patients undergoing invasive coronary. -> invasive coronary angiography?
Reviewer 2 Report
The authors sought to perform a retrospective analysis of patients who had Intra aortic balloon pump for peri procedural support in cardiac cath lab, and they evaluated characteristics that will predict in hospital mortality. They report high rates on mortality in those who had IABP support.
- Introduction should be more concise and suggest the premise for the topic.
- The results are majorly subject to confounding. It is unclear if all pertinent factors were adjusted for in the mortality analysis.
- What was the degree of shock? Any markers of end organ dysfunction such as lactate, Cr, LFTs? Was it guided by RHC measures of CO?
Round 2
Reviewer 1 Report
The authors sufficiently responded to the reviewers’ comments. Now the paper can be accepted.